# Systematic Analysis of the Grafting-Related Glucanase-Encoding GH9 Family Genes in Pepper, Tomato and Tobacco

**DOI:** 10.3390/plants11162092

**Published:** 2022-08-11

**Authors:** Guangbao Luo, Xinran Huang, Jiawei Chen, Jinying Luo, Yufei Liu, Yunfei Tang, Mu Xiong, Yongen Lu, Yuan Huang, Bo Ouyang

**Affiliations:** 1Key Laboratory of Horticultural Plant Biology (Ministry of Education), Huazhong Agricultural University, Wuhan 430070, China; 2Shenzhen Institute of Nutrition and Health, Huazhong Agricultural University, Shenzhen 518120, China; 3Shenzhen Branch, Guangdong Laboratory for Lingnan Modern Agriculture, Genome Analysis Laboratory of the Ministry of Agriculture, Agricultural Genomics Institute at Shenzhen, Chinese Academy of Agricultural Sciences, Shenzhen 518120, China

**Keywords:** pepper, grafting, GH9 family, conserved motif, wound responsive, auxin responsive, gene expression

## Abstract

Grafting is an important agricultural practice to control soil-borne diseases, alleviate continuous cropping problems and improve stress tolerance in vegetable industry, but it is relatively less applied in pepper production. A recent study has revealed the key roles of β-1, 4-glucanase in graft survival. We speculated that the GH9 family gene encoding glucanase may be involved in the obstacles of pepper grafting. Therefore, we performed a systematic analysis of the GH9 family in pepper, tomato and tobacco. A total of 25, 24 and 42 GH9 genes were identified from these three species. Compared with the orthologues of other solanaceous crops, the deduced pepper GH9B3 protein lacks a conserved motif (Motif 5). Promoter cis-element analysis revealed that a wound-responsive element exists in the promoter of tobacco *NbGH9B3*, but it is absent in the GH9B3 promoter of most solanaceous crops. The auxin-responsive related element is absent in *CaGH9B3* promoter, but it presents in the promoter of tobacco, tomato, potato and petunia GH9B3. Tissue and induction expression profiles indicated that GH9 family genes are functionally differentiated. Nine GH9 genes, including *CaGH9B3*, were detected expressing in pepper stem. The expression patterns of *NbGH9B3* and *CaGH9B3* in grafting were different in our test condition, with obvious induction in tobacco but repression in pepper. Furthermore, weighted correlation network analysis (WGCNA) revealed 58 transcription factor genes highly co-expressed with *NbGH9B3*. Eight WRKY binding sites were detected in the promoter of *NbGH9B3*, and several *NbWRKYs* were highly co-expressed with *NbGH9B3*. In conclusion, the missing of Motif 5 in CaGH9B3, and lacking of wound- and auxin-responsive elements in the gene promoter are the potential causes of grafting-related problems in pepper. *WRKY* family transcription factors could be important regulator of *NbGH9B3* in tobacco grafting. Our analysis points out the putative regulators of *NbGH9B3*, which would be helpful to the functional validation and the study of signal pathways related to grafting in the future.

## 1. Introduction

Grafting is the process in which different parts of a plant fuse and establish vascular connections under natural or artificial conditions to form a single plant [1]. Application of grafts in Asia and Europe has been documented in Greek and Chinese literature dating back to the 5th century BC [2,3]. Grafting technology is widely used in modern agriculture. It plays an important role in vegetative propagation, shortening juvenility of fruit trees, renewing varieties, improving nutrient absorption, regulating ion accumulation and enhancing resistance [4,5,6]. Meanwhile, grafting is also an important tool for fundamental research, especially for studying long-distance signaling [1,3,7]. With the aid of grafting, a variety of macro and small molecules have been identified as mobile signals, which are transmitted in the vascular system to regulate specific physiological and developmental processes [8]. Grafting is also used in the study of systematic signaling in plants [9]. Previous application of grafting in horticulture mainly focuses on fruit trees. However, it is now extensively used on vegetable crops worldwide [10]. Typically, grafting is used to increase the resistance of vegetables to soil-borne diseases and nematodes, and to improve yield and fruit quality [11,12,13].

Although grafting technology plays a vital role in agricultural practice and facilitates basic research, the information is rather limited regarding the mechanism of graft body formation and graft incompatibility. Potential factors affecting graft compatibility include cell–cell recognition, wound response, growth regulators, peroxidase and incompatibility toxicity [14,15]. The functional reconnection of the vascular system is key for scion survival [1,16]. Previous studies have shown that callus formation at the graft interface will generate the secondary xylem and phloem, and then develop into a new vascular reconnection [17]. However, a recent study of self-grafting and heterologous grafting between tobacco and Arabidopsis shows that vascular reconnection is independent of the secondary phloem or xylem differentiated by callus [18]. In this study, three phases can be observed in the graft union: (1) the rapid segmentation of long vascular into small and short tracheary elements, (2) overlapping and fusion occur with the accumulation of tracheary elements and (3) the head-to-head connection forms a coherent vascular system. These three phases achieve compatible grafting, while in incompatible grafting, no vascular reconnection occurs even if the tracheary elements are close enough and overlapped. This is probably because the tracheary elements form in an incompatible grafting are spiral-like and hard to reconnect.

Considerable efforts have been made to identify genes involved in the signaling and vascular development in grafting, which is beneficial to the elucidation of the mechanism underlying grafting incompatibility and vascular reconnection. Hormone-related mutant analysis demonstrated that auxin, rather than cytokinin, plays critical roles in phloem reconnection in grafting [19]. Auxin also induces the differentiation of vascular tissue, and accelerates the healing of grafting wounds [16,20]. Gene network regulating graft formation has also been investigated in two recent studies, which have discovered several candidate hub genes during graft formation and validated the critical roles of *SlWOX4* in tomato grafting [21,22]. Nevertheless, the molecular mechanism of grafting healing remains to be further studied. Here, we focused on exploring the potential reasons underlying the problem of pepper grafting.

Pepper (*Capsicum annuum*) is the most important condiment. It also serves as important raw materials for the cosmetic and pharmaceutical industry. In the past two decades, global pepper production has increased from 17 to 36 million tons, and the cultivation area has expanded by 35% [23]. However, pepper production is being challenged by continuous cropping obstacles and the ever-increasing biotic and abiotic stresses [12,24]. As grafting can effectively solve the above problems in vegetable production [11,13], it should be employed in the pepper industry. Nevertheless, the application of grafting on pepper is relatively less often when compared with other major vegetable crops, such as watermelon, melon and tomato [25,26]. There could be at least two reasons: (1) rootstock breeding in pepper is lagging behind [26]; (2) pepper grafting is more challenging than that in other species mentioned above, especially in the heterograft of pepper/tomato (scion/stock) [22,27]. Practice from interspecific grafting among solanaceous vegetables in previous studies [22,27,28], our colleagues and ourselves revealed a lower survival rate of grafts when pepper was used either as rootstock or as scion. We speculate that this issue is likely related to the β-1,4-glucanase (GH9) family, as the previous study has proved that GH9B3 is a vital regulator of interfamily grafts, and its upregulation contributes to graft survival among 80 species [29].

The GH9 family genes in plants can be divided into three subfamilies. The GH9A subfamily is extremely important in cellulose assembly, and its member comprises a transmembrane domain, a catalytic structure and an N-terminal extension [30]. The protein of GH9B subfamily is secretory, with a catalytic structure and signal peptide [31]. Members of the GH9C subfamily can bind to crystalline cellulose, which possess a CBM49 domain at the C-terminus [32]. GH9C proteins can hydrolyze a variety of polysaccharides in plant cell walls [33]. Recently, *NbGH9B3*, a β-1,4-glucanase-encoding gene from tobacco, has been characterized to be an important genetic regulator for graft survival in interfamily grafting [29]. Tobacco can be used as an interscion to achieve Arabidopsis-tomato and chrysanthemum-tomato grafting, and the success of interfamily grafting is dependent on the expression of *NbGH9B3*. Overexpression of *NbGH9B3* in Arabidopsis improves grafting survival rate significantly, while knockout of this gene impairs grafting survival. This important discovery sheds light on addressing important issues regarding grafting in solanaceous vegetables, especially in chili pepper.

Here, it is speculated that GH9 family may attribute to the grafting problem of pepper, and in order to study the putative regulatory factors of GH9B3, we systematically analyzed the GH9 members in pepper, tobacco and tomato, focusing on the comparison of the functionally proved *NbGH9B3*, its orthologue, co-expression network and putative regulators.

## 2. Results

### 2.1. The GH9 Family Members in Tomato, Pepper and Tobacco

After searching and confirming, 25, 24 and 42 GH9 family members were identified in the genome of pepper, tomato and tobacco, respectively (Appendix A). Based on the orthologue relationship with Arabidopsis GH9 members, the GH9 genes were renamed for the three solanaceous crops. In pepper, more GH9 genes were distributed on chromosome 5 and 1, with five and four members, respectively. In tomato, there were four GH9 members on chromosome 8, and most other chromosomes owned two GH9 genes, except for chromosome 6 and 10. In tobacco, the 42 GH9 members were distributed over 37 genomic scaffolds.

### 2.2. Phylogenetic Relationship of GH9 Family Members

In order to understand the molecular evolutionary relationship of GH9 family members in the three solanaceous crops, we constructed an maximum-likelihood tree for all the members identified (Figure 1). Overall, the GH9 family members can be divided into three clades (Clade I, II, and III), which mainly include members of the subfamily GH9A, GH9B and GH9C, respectively. The GH9B3 genes of these three species are closely grouped in a small branch in Clade II.

### 2.3. Gene Collinearity and Duplication in GH9 Family

Gene collinearity and duplication relationship can provide evidence about the functional relationship of GH9 family members within and among the species. Here, MCSCAN2 was used to investigate the collinearity and replication events of the GH9 family members in pepper. Moreover, OrthoFinder was used to analyze the duplication events and orthologous relationship of GH9 members among pepper, tomato and tobacco. No tandem duplication was detected in the pepper genome. The GH9 member genes with collinearity in pepper are *CaGH9A2b* and *CaGH9C3*, *CaGH9B18* and *CaGH9B7*, *CaGH9C1* and *CaGH9B8*, *CaGH9B11* and *CaGH9B16*, *CaGH9B5* and *CaGH9B7* and *CaGH9A1* and *CaGH9A3*. There is no collinearity between *CaGH9B3* and other GH9 members (Figure 2). Similarly, no duplication event was detected for *SlGH9B3* and *NbGH9B3* within their genomes.

### 2.4. Gene Structure of GH9 Members

As gene structure provides fundamental information about the transcription of a gene, investigating the structure of GH9 family can give us some basic knowledge into the function of GH9 family members. The gene structure of the GH9 family members was based on the annotations of the three reference genomes (Figure 3A; Appendix A). In pepper, the number of exons in GH9 genes varies from one (*CaGH9B6*) to twelve (*CaGH9C1*). In tomato, the most exons were detected in *SlGH9C1*, the orthologue of *CaGH9C1*, while only two exons were detected in three SlGH9 members, *SlGH9A3*, *SlGH9B4* and *SlGH9C3b*. As for tobacco, the *CaGH9C1* orthologue, *NbGH9C1*, also has the largest number of exons (11), but *NbGHA1b* has only one exon. For GH9B3, both *CaGH9B3* and *SlGH9B3* contain five exons, while *NbGH9B3* has six exons.

### 2.5. Conserved Domains and Motif Distribution of GH9 Members

Conserved domains are key modules of protein function. Domain prediction results showed that all GH9 members contain the conserved domain Glyco_hydro_9, but the domain length and amino acid composition are diverse to some extent. Moreover, members of the GH9C subclass possess an additional CBM49 domain (Appendix A). The results of domain analysis prompted us to further investigate the conserved motifs in GH9 family members. A total of 20 conserved motifs were detected (Figure 3B and Appendix A). Interestingly, one motif (Motif 5) is missing in CaGH9B3 but not for NbGH9B3 and SlGH9B3. We further analyzed the conservation of the motifs of GH9B3 orthologues in six solanaceous crops (tomato, pepper, tobacco, eggplant, petunia, potato) and Arabidopsis. The results confirmed the missing of Motif 5 in pepper GH9B3, as compared to GH9B3 from the other species investigated (Figure 4A).

### 2.6. Prediction of GH9B3 Protein Structure

The biological function of a protein depends on the three-dimensional (3D) structure of the protein. Although the results of motif analysis showed that CaGH9B3 is different from its orthologues in other species tested, little is known about the 3D structure of GH9B3 in different species. Therefore, we employed RoseTTAFold to predict the 3D structure (Figure 4B). Overall, the 3D structure of GH9B3 from these three species looks similar. All GH9B3 in the three species harbor a structure of six sheets. Nevertheless, there is a clear difference between CaGH9B3 and SlGH9B3/NbGH9B3 regarding helix and loop structure. NbGH9B3, SlGH9B3 and CaGH9B3 contain 14, 12 and 10 helixes, respectively. A long helix (black arrow 1 in Figure 4B) at the N terminal of NbGH9B3 could be detected and a shorter helix can be observed at the same position in SlGH9B3, but no helix is observed in CaGH9B3 at that position. Two small helixes (black arrow 2 and 3) can be found in tomato and tobacco GH9B3, but were missing in pepper GH9B3. In addition, the loop structure of pepper GH9B3 is also different (e.g., the part in the green rectangle of Figure 4B) from that of tomato and tobacco.

### 2.7. Distribution of Cis-Acting Elements in the Promoter of GH9 Family Genes

As cis-acting elements play critical roles in gene regulation, the 2000 bp promoter region of all the GH9 family genes from these three species were used to predict the cis-elements (Appendix A). It was found that cis-elements related to gibberellin (GA), abscisic acid (ABA), salicylic acid (SA), zeatin, jasmonic acid (JA), light, low temperature and cell cycle responsive are frequently detected (Figure 5). We have noticed some difference regarding wound- and auxin-responsive element in the promoter of GH9B3 genes. Further analysis revealed this difference applies to the GH9B3 promoters of main solanaceous crops (tomato, petunia, potato, tobacco and pepper) (Appendix A). A wound-response element was detected in the promoter of tobacco *NbGH9B3*, but this element cannot be detected in the GH9B3 promoter of other solanaceous crops. Meanwhile, auxin-related elements were detected in the promoter of tobacco, tomato, potato and petunia GH9B3 genes, but it was missing in the GH9B3 promoters of the other solanaceous crops investigated (Appendix A).

### 2.8. Tissue Expression Profile of Pepper GH9 Family Genes

To gain clues to the function of GH9 family genes in pepper and characterize the expression pattern of GH9 family members, we retrieved the expression data from PepperHub and combined the transcriptome data of stem to obtain the tissue expression profile. Among the 25 pepper GH9 family genes, 14 are expressed in leaves, flower organs and fruit tissues, and nine were expressed in stem (Figure 6). *CaGHC3* and *CaGHA1* are highly expressed during leaf development, and eight GH9 genes are expressed actively during flower development, such as *CaGH9A1*, *CaGH9C3*, *CaGH9B13*, etc. In fruit, the expression level of *CaGH9C3*, *CaGH9A1*, *CaGH9B11*, *CaGH9B5*, *CaGH9B8* and *CaGH9B18* is high in the early and mid-term stages, while the expression level of *CaGH9B3* and *CaGH9B15* is high in the later period of fruit development. *CaGH9C2*, *CaGH9A2a*, *CaGH9B3*, *CaGH9B15* and *CaGH9C3* are actively expressed during placenta development. The expression of *CaGH9B5*, *CaGH9B8* and *CaGH9B11*, is relatively high in the early stages of seed development. As for stem tissue, active expression of *CaGH9B18*, *CaGH9C3* and *CaGH9B11* was observed, while *CaGH9B3* showed a moderate expression level. The above results indicate that these genes may be critical for pepper growth and development in a tissue-specific way.

### 2.9. Induction Expression Profiles of GH9 Family Genes in Pepper

The results of cis-element analysis showed that there are many phytohormone- and stress-responsive elements detected in the promoter of pepper GH9 genes. Herein, we investigated their expression profiles under different phytohormone and stress treatments. It was found that the expression of pepper GH9 genes is affected by SA, ABA, GA, IAA or JA (Figure 7A). Among them, *CaGH9C2*, *CaGH9A3*, *CaGH9B5*, *CaGH9A2a* and *CaGH9B11* are induced by SA and *CaGH9B13*, *CaGH9B18*, *CaGH9B14*, *CaGH9C3* and *CaGH9A2b* are suppressed by ABA. *CaGH9A4* is induced by JA, IAA and GA, and *CaGH9B3* is induced by SA and GA. These results indicate that GH9 family in pepper shows diversity in hormone responsiveness, which is consistent with the prediction results of cis-element.

The gene expression data of stress treatment showed that the expression of 19 GH9 genes changes under different stresses. *CaGH9B7* is significantly upregulated by H_2_O_2_ treatment, while *CaGH9A2b*, *CaGH9B14*, *CaGH9B15*, *CaGH9B3*, *CaGH9B5*, *CaGH9B18*, *CaGH9C3* and *CaGH9B13* are repressed by this treatment. *CaGH9A3*, *CaGH9B11* and *CaGH9C2* are induced by low temperature, while other genes do not show a dynamic expression pattern. *CaGH9B15* is induced by heat stress, but the rest do not respond to heat stress. *CaGH9A1*, *CaGH9B6* and *CaGH9A2b* are induced by NaCl treatment (Figure 7B). These results indicated that GH9 family members may play multiple roles in response to different stress.

### 2.10. Co-Expressed Transcription Factor Genes

Weighted correlation network analysis (WGCNA) is a powerful method to identify target genes, biomarkers and regulatory networks [34,35,36].To identify putative regulators of GH9B3, we conducted WGCNA analyses on the dynamic transcriptome data of tobacco grafting experiment and the large-scale transcriptome data of pepper development. Results from tobacco showed that there are 524 genes having a weighted value more than 0.2 with *NbGH9B3*, of which 58 encode transcription factors (Figure 8; Appendix A). The 58 co-expressed transcription factors can be divided into two clusters. Most transcription factor genes have a high expression level at 2 h, but the 19 transcription factor genes closely clustered with *NbGH9B3* have a low expression level at this time point. From day 1 to day 28, the 19 transcription factor genes showed a highly similar expression pattern with *NbGH9B3*. Three WRKY family genes (*Niben101Scf00996g04023.1*, *Niben101Scf01507g00004.1* and *Niben101Scf05584g00005.1*) have the highest co-expression weighted value with *NbGH9B3*. In addition, a gene encoding AUX/IAA regulator (*Niben101Scf01218g00007.1*) and five AP2/ERF family genes (*Niben101Scf03548g02037.1*, *Niben101Scf01142g04009.1*, *Niben101Scf00428g09009.1*, *Niben101Scf08546g05002.1* and *Niben101Scf00163g22002.1*) also have a relatively higher weighted value with *NbGH9B3*.

### 2.11. Distinct Expression Pattern of GH9B3 in Pepper and Tobacco Self-Grafting

Notaguchi et al., (2020) has demonstrated that tobacco *NbGH9B3* plays a key role in the survival rate of interfamily grafting. Our analysis of cis-elements showed that there was no wound-responsive element in the pepper GH9 family genes expressed in stem, including *CaGH9B3*. However, a wound-responsive element can be detected in the promoter of tobacco *NbGH9B3*. In addition, an auxin-responsive element was detected in the promoter of *NbGH9B3* but not in that of *CaGH9B3* (Appendix A). It is worth noting that in a recent study on the heterograft of tomato and pepper, no regulator of *CaGH9B3* was detected [22]. In contrast, 58 transcription factor genes were detected to be highly co-expressed with *NbGH9B3* (Appendix A). To investigate the potential effect of these differences, we performed self-grafting in pepper and tobacco to mock a wound treatment, and detected the expression pattern of GH9B3. Distinct expression patterns were identified for *NbGH9B3* and *CaGH9B3*. The expression level of *NbGH9B3* in tobacco scion was increased significantly 24 h post grafting, with a 14-fold increase in expression. However, the expression level of *CaGH9B3* in pepper scion was decreased to half of that before treatment, and further decreased to 5% after 24 h (Figure 9).

## 3. Discussion

### 3.1. Members, Phylogenetic Relationship and Duplication of GH9 Family Genes

In this study, we identified 25 GH9 members in pepper genome. Among them, 5, 17 and 3 members belong to GH9A, GH9B and GH9C subfamily, respectively. In Arabidopsis, the corresponding numbers are 4, 18 and 3. In tomato, 24 GH9 members were identified, with 5, 14 and 5 members in the three subfamilies, respectively. The length of GH9 genes in pepper ranges from 377 bp to 6570 bp, and it ranges from 342 bp to 6748 bp in tomato, which is similar to that of pepper. The total numbers of GH9 members in pepper and tomato is close to that of most reported species, including barley, rice, sorghum, poplar and Arabidopsis [37,38,39]. However, 42 GH9 genes were identified in tobacco, with 14, 20 and 8 members in the three subfamilies. The main reason for the higher number is that tobacco is an allotetraploid species [40]. A total of 12 duplication events could be detected in the tobacco GH9 family (Appendix A). Interestingly, the duplication event of GH9B3 was not detected within the tomato, pepper or tobacco genome.

### 3.2. The Difference in Gene Structure and Conserved Motif

Gene structure helps to infer homology and here we found the coding sequences of GH9 members of tobacco, tomato and pepper are similar in structure to their homologous members from rice and maize [37,39]. Genes on the same phylogenetic branch have similar numbers of exons and introns. Pepper and tomato GH9B3 are one exon less than tobacco GH9B3. Interestingly, regarding conserved motif, no difference was detected in GH9B3 from tomato, tobacco, potato, petunia, eggplant and Arabidopsis. However, one motif (Motif 5) was missing in pepper (Figure 4A). The 3D structure of protein also showed that pepper GH9B3 has clear differences in spatial structure compared with tobacco and tomato GH9B3. In view that the grafting of pepper is more difficult in the mutual grafting of pepper, tobacco and tomato, we speculate that the lack of Motif 5 may affect the function of CaGH9B3. This hypothesis can be tested by deleting the specific motif in transformation-friendly species, such as tomato and tobacco.

### 3.3. Cis-Elements in GH9 Family Promoter and the Expression Profiles

The diversity of cis-elements in promoter is of great significance to pleiotropy of a gene [41]. The prediction results showed that the promoters of GH9 family members in tomato, tobacco and pepper are enriched with cis-elements related to various phytohormones, stresses, light and cell cycle response. These results suggested that GH9 family members would have a complex expression regulatory network in plant development, stress and phytohormone response. Consistent with this, the expression of GH9 family genes in pepper is changed upon various phytohormones (Figure 7A) and stresses (Figure 7B). Previous studies have documented that auxin is essential for the reconnection of vascular tissues [16,20]. Based on our data, for the CaGH9B members expressed in pepper stem, *CaGH9B3* (the promoter contains no auxin responsive element) is not induced by IAA treatment, which may contribute to the obstacle of pepper grafting. Moreover, a previous study has revealed that ABA is essential for stem scar healing in tomato [42], while the expression profile shows that *CaGH9B13*, *CaGH9B18*, *CaGH9B14*, *CaGH9C3* and *CaGH9A2b* are repressed by ABA treatment.

Until now, little is known about the stress response of the GH9 family genes. Here, we found that the GH9 family genes display diverse expression patterns under different stress treatments. For instance, only *CaGH9A3* and *CaGH9C2* are dramatically induced by cold treatment, which implies that these two genes may be important for cold defense. No gene is upregulated by NaCl treatment, and only *CaGH9B15* is upregulated by heat treatment. These findings may help us to further understand the roles of GH9 family genes in stress conditions.

Hydrogen peroxide, as an important signal molecule, directly regulates the expression of many genes in plants, which is crucial in the process of programmed cell death, defense response and environmental adaptation. In the treatment of H_2_O_2_, we found that approximately ten GH9 genes are downregulated in pepper. Interestingly, *CaGH9B7* showed an upregulation pattern, which implies a particular function of GH9 family under the H_2_O_2_ condition. Further experiments on this gene will help us to reveal its roles.

Nine members of GH9 are detected expressing in pepper stem, but no wound-responsive element was predicted in the promoter of these genes. In rice, the GH9 members *OsGH9B5*, *OsGH9B8*, *OsGH9B9*, *OsGH9B10* and *OsGH9B11* are co-expressed in cell wall [39]. Rice *OsGH9B1/2/3/16* and Arabidopsis *AtGH9B1/2* play an enzymatic role in the transformation of lignocellulose crystallinity, and may have specific activity in the post-modification of cellulose [39]. We found that *CaGH9B11*, *CaGH9B5*, *CaGH9B8* and *CaGH9B3* are expressed in pepper stem. These GH9 members could be important for the cell wall behaviors in stem tissue.

Recently, the critical role of *NbGH9B3* in improving interfamily graft survival was found [29]. In addition, GH9B3 is significantly upregulated in self-grafted soybean, Arabidopsis and maize, which implies a conserved role of GH9B3 in plant cell-cell adhesion and graft survival [29]. Our analysis of the cis-elements in GH9B3 promoter of solanaceous crops (tomato, pepper, tobacco, petunia, eggplant and potato) showed that the auxin-responsive related element could only be detected in the promoter of tobacco, tomato, petunia and potato GH9B3 (Appendix A). The expression data also showed that *CaGH9B3* is not induced by IAA. Meanwhile, there is a wound-responsive element detected in the GH9B3 promoter of tobacco, but not in the other five solanaceous crops investigated. This element might be a contributing factor for the strong ability of tobacco in interfamily grafting, while a recent study documented that heterograft of pepper/tomato or tomato/pepper displays a dramatic delayed wound response which leads to grafting incompatibility [22]. The distinct gene expression patterns between tobacco and pepper also supported this speculation (Figure 9). This hypothesis can be further tested by grafting experiment using transgenic lines with *SlGH9B3* driven by the tomato GH9B3 promoter incorporated with the wound-responsive element. Nevertheless, the results from tobacco showed there are 58 transcription factor genes highly co-expressed with *NbGH9B3*. Meanwhile, regulators of GH9B3 could be detected neither in pepper/tomato nor tomato/pepper heterograft [22]. The *NbWRKY* (*Niben101Scf00996g04023.1*) showed an expression pattern that was strikingly similar to *NbGH9B3*. Meanwhile, there were eight WRKY binding sites detected on *NbGH9B3* promoter, suggesting that this WRKY gene could be a key regulator of *NbGH9B3* during grafting.

In this work, we have found some major differences on promoter cis-element and protein motif regarding GH9B3, which can be further tested by Agrobacterium-based transformation-friendly species, such as tobacco and tomato. Meanwhile, natural variation of this gene can be explored in the golden age of genomics. In addition, putative regulators of GH9B3 revealed here are promising targets in future studies to deepen our understanding of plant grafting. Specific GH9 members with interesting expression patterns under stress conditions are also good targets for functional research.

## 4. Materials and Methods

### 4.1. Identification of GH9 Family Members and Phylogenetic Analysis

The reference genome of pepper (*Capsicum annuum*, CM1.55), tomato (*Solanum lycopersicum*, SL4.0) and tobacco (*Nicotiana benthamiana*, Niben101) was downloaded from the Sol Genomics Network website (SGN, https://solgenomics.net/ (accessed on 2 July 2021)). The Hidden Markov Model (HMM) of Glycosyl hydrolase family 9 (GH9) was retrieved from the Pfam database [43]. The protein sequences of candidate GH9 family members from the three species were searched using HMMER3.3 software with default parameters [44]. The putative member was further confirmed by BLAST search [45] and NCBI Conserved Domain Database (CDD) analysis (https://www.ncbi.nlm.nih.gov/Structure/cdd/cdd.shtml/ (accessed on 10 June 2021)).

MEGA (https://www.megasoftware.net, Ver7 (accessed on 10 October 2019)) was used to construct the maximum-likelihood tree with 1000 bootstrap replicates [46]. The phylogenetic tree was visualized using the iTOL web server [47].

### 4.2. Collinearity and Duplication Analysis of GH9 Family

Collinearity analysis was performed using MCSCAN2 (http://chibba.pgml.uga.edu/mcscan2/ (accessed on 6 July 2021)) with default parameters [48]. The R package Circlize [49] was used to visualize the relationship. The orthologue relationship and duplication events were analyzed with OrthoFinder [50].

### 4.3. Gene Structure, Motif Discovery and Conserved Domain Scan

Gene structure analysis was based on the annotations of the reference genomes. Motif discovery was conducted at the MEME website (https://meme-suite.org, Ver4.12.0/ (accessed on 6 July 2021)) with classic mode parameters [51]. NCBI CDD website (https://www.ncbi.nlm.nih.gov/Structure/cdd/wrpsb.cgi/ (accessed on 6 July 2021)) and HMMSCAN program from HMMER3 were used to confirm the domain. The results were visualized through TBtools [52].

### 4.4. Prediction of the Three-Dimensional Structure for GH9B3 Orthologues

RoseTTAFold (https://github.com/RosettaCommons/RoseTTAFold/ (accessed on 9 August 2021)) was used to predict the three-dimensional structure of the deduced GH9B3 protein from pepper, tomato and tobacco [53]. Parameters (-mact = 0.35, -maxfilt = 100,000,000, -neffmax = 20, -cov = 25, -maxseq = 1,000,000) were used to generate the msa file. Secondary structures were predicted using the PSIPRED with default parameters within the RoseTTAFold software. To search for the templates, the HHsearch program with parameters (-e 100, -p 5.0, -mact 0.05) were used [54].

### 4.5. Prediction of Cis-Element in the Promoter of GH9 Genes

The 2000 bp upstream sequence of all the genes was used as input to predict the cis-acting regulatory elements in the PLANTCARE database [55], and TBtools was used to visualize the results [52].

### 4.6. Expression Profiling of the Pepper GH9 Family Members

The transcriptome data of different tissues (leaf, flower, pericarp, placenta and seed) and from stress and phytohormone treatments were retrieved from PepperHub [56]. Samples of pepper stem tissue for RNA isolation were taken from 15-day-old seedlings with three biological replicates. And the seedlings were grown in a chamber with the light intensity of 3500 Lux, temperature at 24–26 °C and a photoperiod of 16 h light/8 h darkness. RNA library construction and sequencing was performed at Novogen Ltd. (Tianjin, China). FastQC with default parameters was used to evaluate the quality of sequencing data [57]. After removing the adapter and low quality reads, clean reads were mapped to the reference genome of pepper (CM1.55) using HISAT2 [58]. FeatureCounts was then used to calculate the abundance of transcript [59]. Cluster analysis of the expression data was conducted after log2 transformation of RPKM values through TBtools.

### 4.7. WGCNA Analysis of Tobacco

The tobacco transcriptome data from a grafting experiment were downloaded from EBI (https://www.ebi.ac.uk/ (accessed on 25 March 2022)) under the accession number DRA009936. Genome and gene models’ annotation file of tobacco were released from the Sol Genomics Network website (https://solgenomics.net/ (accessed on 3 January 2021)). We applied the R package WGCNA to conduct weighted gene co-expression network [60]. RPKM data were submitted to log2 transformation and data were preprocessed according to the FAQ of WGCNA (https://horvath.genetics.ucla.edu/html/CoexpressionNetwork/Rpackages/WGCNA/faq.html/ (accessed on 11 October 2021)). Step-by-step network construction was performed by following the tutorials (https://horvath.genetics.ucla.edu/html/CoexpressionNetwork/Rpackages/WGCNA/Tutorials/index.html/ (accessed on 11 October 2021)).

### 4.8. Expression Analysis of GH9B3 in Pepper and Tobacco in Grafting

Pepper (*Capsicum annum*, cv. MiniPep) and tobacco (*Nicotiana benthamiana*) seedlings were grown under conditions as mentioned above. Standard splice self-grafting was performed when the seedlings grew to 5–6 true leaves. Samples were taken before (0 h) and after (6 h, 24 h) cutting and grafting, and approximate 3 cm scion stems from the cut were collected for gene expression analysis. RNA was extracted with TRIzol (Invitrogen, Carlsbad, CA, USA) and reverse transcribed with HIScriptII reagent (Vazyme, Nanjing, China) according to the instructions. The relative expression level of GH9B3 gene to a housekeeping gene was detected by Roche qRT-PCR with LightCycler 96 (Roche, Basel, Switzerland). The reaction volume was 10 μL, containing 5 μL of SYBR premix, 0.5 μL of forward and reverse primer each and 4 μL cDNA template. The cycling parameters are: 95 °C for 5 min; 40 cycles of 95 °C for 10 s, 60 °C for 10 s and 72 °C for 20 s. Tobacco *EF-lα* (GenBank accession no. AF120093.1) and pepper *UBI-3* (AY486137.1) were used as internal control respectively. The ΔΔCt method was used to analyze the expression data [61]. The primers are listed in Appendix A.

## 5. Conclusions

Previous studies on grafting focus less on the identification of genes related to graft survival. In this study, we conducted a systematic analysis of the GH9 family members in three solanaceous crops, with more focus on GH9B3. Our results suggested the lack of Motif 5 in pepper GH9B3 protein sequence may lead to the functional differentiation between *CaGH9B3* and GH9B3 from the other solanaceous species investigated. Moreover, the lack of wound-responsive element and auxin-related element in *CaGH9B3* promoter may result in the different expression patterns in pepper and tobacco grafts, WRKY may be a key regulator of *NbGH9B3* which can be validated in the future. Large-scale resequencing of pepper germplasm would give us an opportunity to explore natural alleles of *CaGH9B3* and its promoter, thus finding potential accessions carrying the wound-responsive element, auxin-related element and/or Motif 5, which would be beneficial to pepper grafting.

## Figures and Tables

**Figure 1 plants-11-02092-f001:**
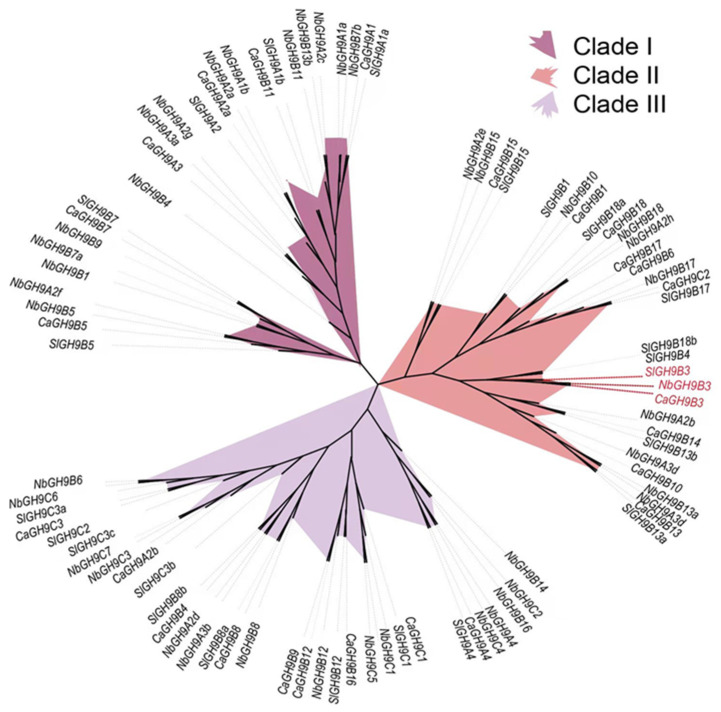
Phylogenetic tree of GH9 family members from pepper (*Capsicum annum*), tomato (*Solanum lycopersicum*) and tobacco (*Nicotiana benthamiana*). MEGA7 was used to construct the maximum-likelihood tree, and the tree was visualized using iTOL web server. Different colors represent different clades, and the red branches are GH9B3.

**Figure 2 plants-11-02092-f002:**
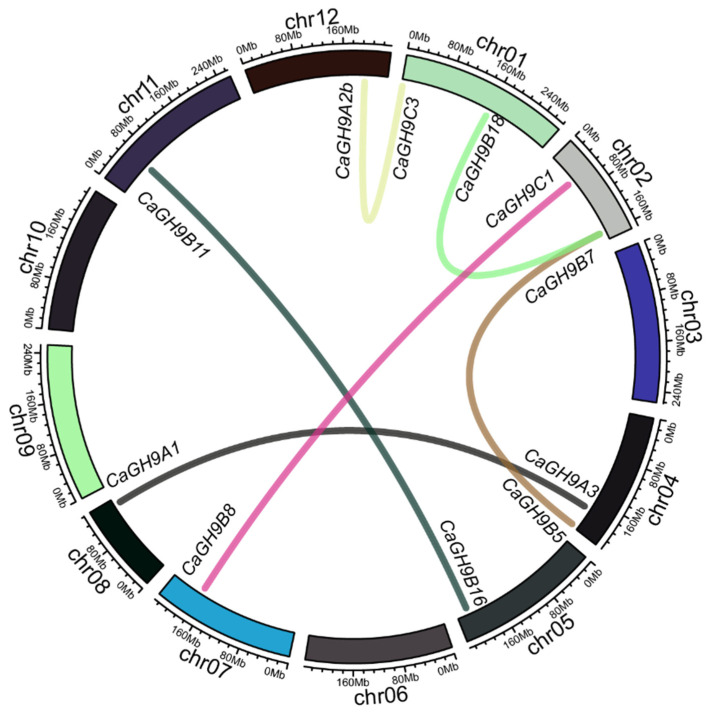
The collinearity among the GH9 family members in pepper. MCSCAN2 was used to calculate the genome-wide collinearity, and the circos plot was performed by Circlize package in R. The reference genome used is CM334 (V1.55).

**Figure 3 plants-11-02092-f003:**
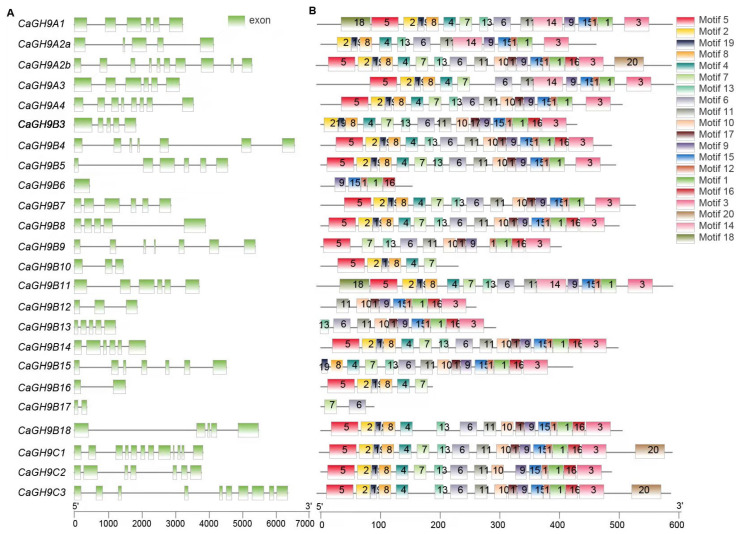
Gene structure and conserved motif of GH9 family members from pepper (*Capsicum*
*annum*). (**A**) Gene structure. Green blocks represent exons and gray lines represent introns; (**B**) Distribution of conserved motifs of GH9 members in pepper.

**Figure 4 plants-11-02092-f004:**
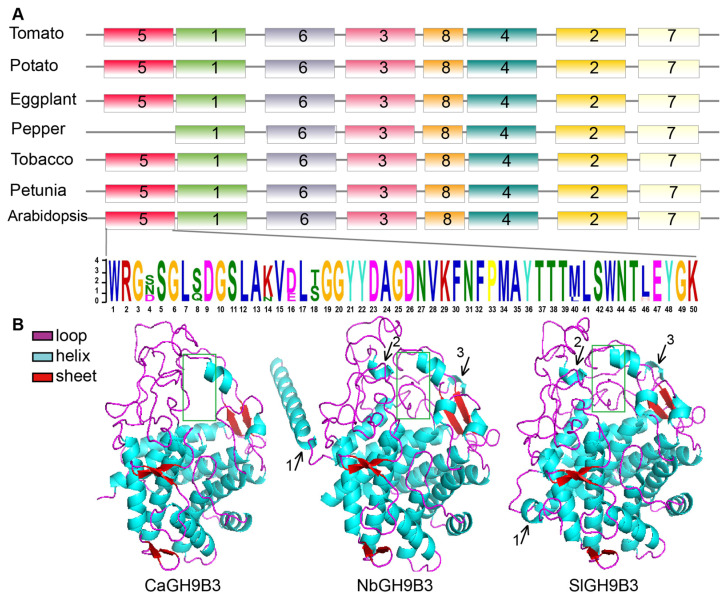
Motif arrangement and protein structure of GH9B3 members. (**A**) Motif 5 is missing in pepper GH9B3. Motif discovery was conducted through the MEME website and seven motifs were detected. (**B**) Protein structure of CaGH9B3, NbGH9B3 and SlGH9B3. The protein structure was predicted through RoseTTAFold. Purple, blue and red cartoons represent loop, helix and sheet structure, respectively. The black arrows (1, 2 and 3) point out the difference of helix structure, and the green rectangle emphasizes the difference of the loop structure from CaGH9B3.

**Figure 5 plants-11-02092-f005:**
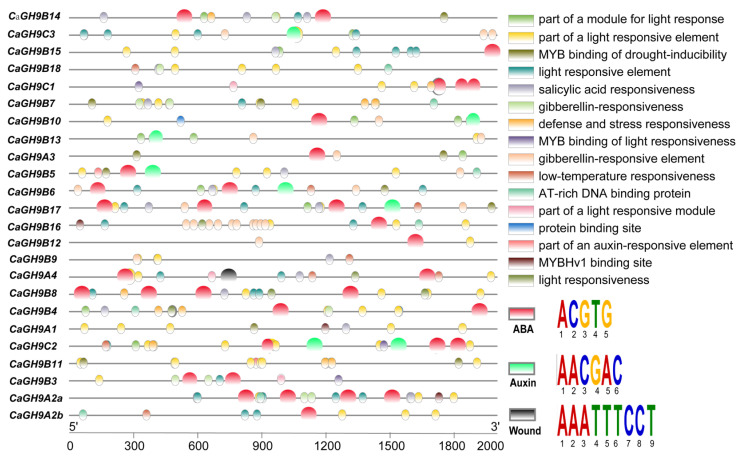
Distribution of cis-elements in the promoter region of GH9 family genes from pepper (*Capsicum annum*). The black, green and red balls represent wound-, auxin- and ABA-responsive element, respectively. These elements are potentially involved in grafting healing, and the wound- and auxin-responsive elements are missing in the promoter of *CaGH9B3*.

**Figure 6 plants-11-02092-f006:**
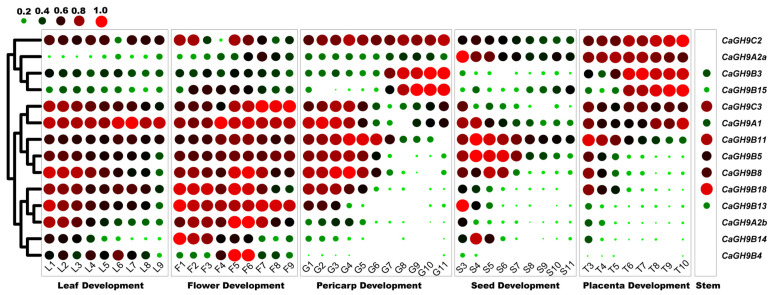
Expression profiles of pepper GH9 family genes in different tissues. Log2 and row scale normalization to 1 were applied to the RNA-seq based reads per kilo base per million mapped reads (RPKM) values to avoid the orders of magnitude difference in gene expression among different GH9 members. RNA-seq data was derived from PepperHub database, and the heatmap visualization was performed using TBtools. The scale (upper left) shows that the expression value gradually increases from small green dot to large red dot. A gene with a 0 RPKM value will have a 0-sized dot.

**Figure 7 plants-11-02092-f007:**
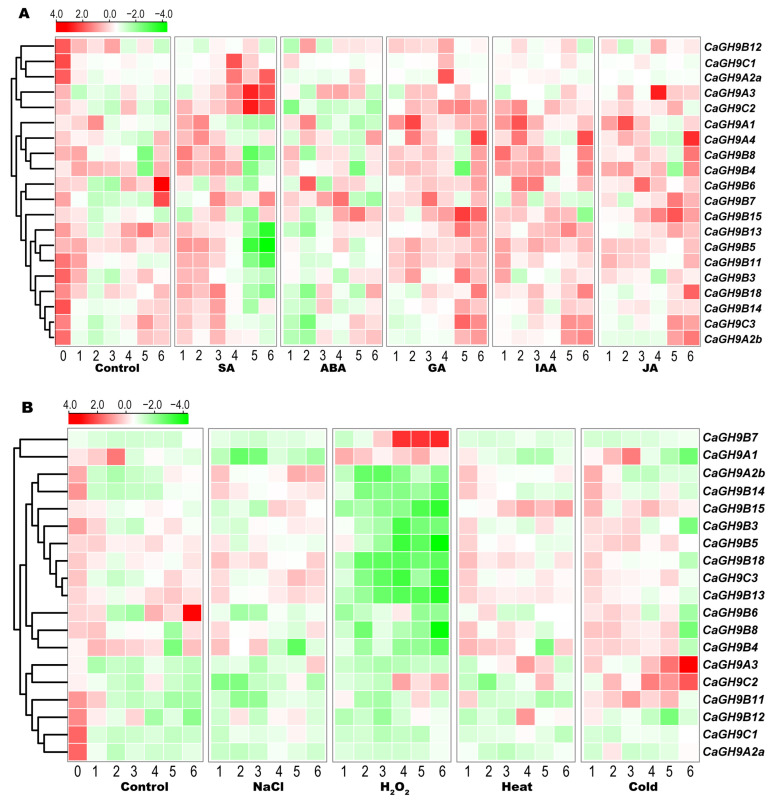
The expression patterns of GH9 family genes under the treatment of phytohormones (**A**) and stresses in pepper leaf (**B**). Log2 and row scale normalization were applied to the RNA-seq based RPKM values, and the heatmap visualization was performed using TBtools. The numbers 0–6 represent different time points (0, 1.5, 3, 6, 12, 24 h) before (0) and after treatment. The color scale shows increasing expression level from green to red. SA, salicylic acid; ABA, abscisic acid; GA, gibberellin; IAA, indole-3-acetic acid; JA, jasmonic acid. RNA-seq data were retrieved from PepperHub.

**Figure 8 plants-11-02092-f008:**
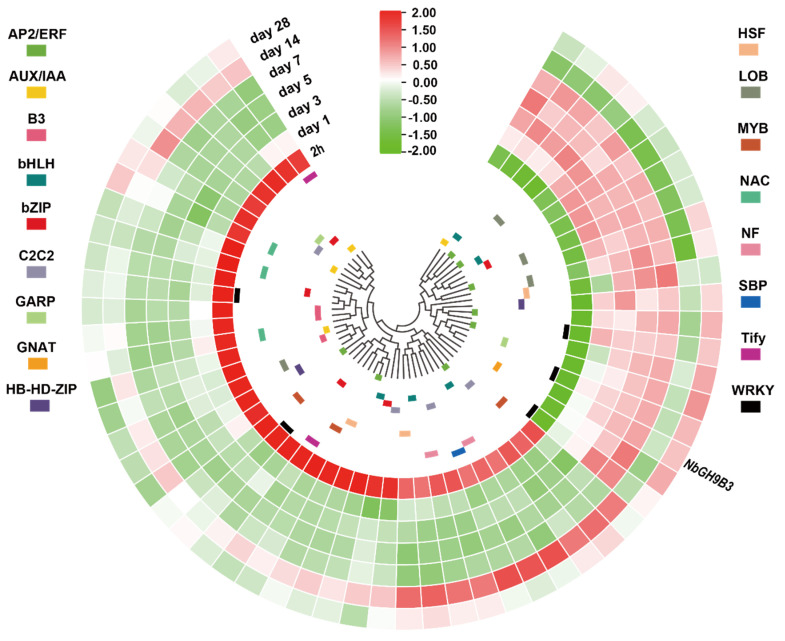
Co-expressed transcription factor genes of *NbGH9B3* in the weighted correlation network analysis (WGCNA) results. Tbtools was used to visualize the expression data. Log2, row scale and normalization was applied on the RNA-seq based reads per kilo base per million mapped reads (RPKM) value. Circle heatmap shows the co-expressed transcription factor genes of *NbGH9B3*.

**Figure 9 plants-11-02092-f009:**
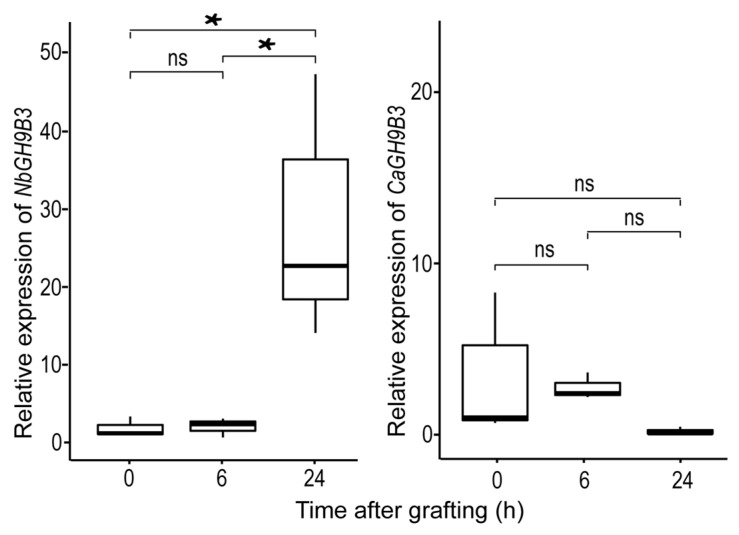
Distinct expression patterns of GH9B3 in pepper and tobacco self-grafting. Quantitative reverse transcription-PCR (qRT-PCR) analysis was performed on the RNA samples extracted from the scion stem segments of pepper and tobacco before (0 h) and after (6 h, 24 h) self-grafting. *UBI-3* and *EF-lα* were used as internal control for pepper and tobacco, respectively. Data shown are means ± SD of three biological replicates. Student’s *t*-test; ns, non-significant; * significant at *p*-value < 0.05.

## Data Availability

The RNAseq data for pepper tissue, stress and phytohormone treatment can be downloaded from PepperHub (pepperhub.hzau.edu.cn). The RNAseq data for tobacco grafting is available from EBI (https://www.ebi.ac.uk/) under the accession number DRA009936.

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
