# Peer review of "Systematic Analysis of the Grafting-Related Glucanase-Encoding GH9 Family Genes in Pepper, Tomato and Tobacco"

_plants, 2022, doi:10.3390/plants11162092_

Round 1

Reviewer 1 Report

The authors have done a great deal of careful work and highlighted the alleged cause of grafting failures in pepper. The role of the GH9B3 gene family in Solanaceae is fairly well covered. The work is innovative and will attract the attention of readers. I believe this work can be accepted by the journal for publication after minor revisions.

Small remarks. In the Abstract section, when describing the CaGH9B3 gene, it is necessary to indicate more clearly where Motif 5 is absent - in the promoter or in the coding sequence, otherwise it will be incomprehensible to readers. The same is in the Conclusion section: Lines 472-473. It would also be desirable to specify more clearly whether it is the absence of motif 5 in the coding region or different cis-motifs in the CaGH9B3 promoter that contributes most to the failure of grafting.

Author Response

1. When describing the CaGH9B3 gene, it is necessary to indicate more clearly where Motif 5 is absent - in the promoter or in the coding sequence, otherwise it will be incomprehensible to readers. The same is in the Conclusion section: Lines 472-473.

Response 1: Thanks for the suggestion. We have clarified in the abstract and conclusion that the Motif 5 is from the deduced protein sequence.

2. It would also be desirable to specify more clearly whether it is the absence of motif 5 in the coding region or different cis-motifs in the CaGH9B3 promoter that contributes most to the failure of grafting.

Response2: Here, we have to compromise that until now, we have no strong evidence to conclude about the contribution of these two differences. As mentioned in the lines 367-368 and 418-420, we need further study to validate the impact of wound response element and Motif 5.

Reviewer 2 Report

 Dear Authors,

I have an opportunity to review manuscript entitled” Systematic Analysis of the Grafting-related Glucanase-encoding GH9 Family Genes in Pepper, Tomato and Tobacco” submitted to Plant MDPI Journal.

Authors concentrated in general on the key roles of β-1, 4-glucanase in graft survival.

A systematic analysis of the GH9 family in pepper, tomato and tobacco were performed; A total of 25, 24, and 42 GH9 genes were identified from these three species. Compared with the orthologues of other solanaceous crops, pepper CaGH9B3 lacks a conserved motif (Motif 5). Interestingly, tissue and induction expression profiles indicated that GH9 family genes are functionally differentiated. Nine pepper GH9 genes, including CaGH9B3, were detected expressing in pepper stem. The ex-pression patterns of NbGH9B3 and CaGH9B3 in grafting were different in our test condition, with obvious induction in tobacco but repression in pepper.

The introduction provide quite sufficient background for the reader. But one aspect should be described in detail – Why Authors chosen exactly that GH9 family as well as speculated that the issue is likely related to the β-1,4-glucanase (GH9) family??

Please, precise the aim of the studies in details;

Moreover, results are in general clear described, but some aspects also need improvements:

-        I suggest to enlarge Figure 3 – fit to the whole page, it is an important aspect and the reader should have clear analyses;

-        On  the other side the expression profile -figure 6 and figure 7- should be rearranged (also enlarged) to the clear analyses- these type of heatmap presentation is completely unreadable! – the legend too; Moreover, “tissue”, phytohormone and stress expression should be more deeply described in results part as well as discussed in discussion section;

-        What about statistical significant changes in Figure 9- it should be completed!

-        Discussion should be completed with future prospects coming from obtained results and maybe also further studies

Author Response

1. The introduction provide quite sufficient background for the reader. But one aspect should be described in detail – Why Authors chosen exactly that GH9 family as well as speculated that the issue is likely related to the β-1,4-glucanase (GH9) family? Please, precise the aim of the studies in details;

Response 1:  Thanks for the suggestion. We have emphasized the aim of this study at line 103-104 in the revised manuscript.

2. I suggest to enlarge Figure 3 – fit to the whole page, it is an important aspect and the reader should have clear analyses;

Response 2: Figure 3 has been enlarged as suggested.

3. On  the other side the expression profile -figure 6 and figure 7- should be rearranged (also enlarged) to the clear analyses- these type of heatmap presentation is completely unreadable! – the legend too; Moreover, “tissue”, phytohormone and stress expression should be more deeply described in results part as well as discussed in discussion section;

Response 3: Sorry for the unclarity. In the revised manuscript, we have improved Figure 6 and legend. For Figure 7, we have changed the heatmap to another style, which would be more readable. Meanwhile, we have described and discussed the results in more detail (please refer to line 266, 276-278 in the result section and lines 384-390, 394-396 in the discussion section).

4. What about statistical significant changes in Figure 9- it should be completed!

Response 4: Thanks for the suggestion. We conducted a Student’s t-test on the data.  Corresponding information has been added to the figure and legend.

5. Discussion should be completed with future prospects coming from obtained results and maybe also further studies

Response 5: Thanks for the suggestion. Future prospects are added at the end of the discussion (L380, 394-396, and 428-434).

Round 2

Reviewer 2 Report

Dear Authors,

I find that almost all of my suggestions were added, despite of weak explanation of choosing β-1,4-glucanase (GH9);

Especially figures quality was improved- e.g. figure 7 -very good reconstruction; as well as statistical analyses information was added.